# Targeted pandemic containment through identifying local contact network bottlenecks

**Shenghao Yang**[1]*, **Priyabrata Senapati**[1], **Di Wang**[2], **Chris T. Bauch**[3], **Kimon Fountoulakis**[1]

**1** School of Computer Science, University of Waterloo, Waterloo, Ontario, Canada, **2** Google Research, Mountain View, California, United States, **3** Department of Applied Mathematics, University of Waterloo, Waterloo, Ontario, Canada

* shenghao.yang@uwaterloo.ca

## Abstract

Decision-making about pandemic mitigation often relies upon simulation modelling. Models of disease transmission through networks of contacts–between individuals or between population centres–are increasingly used for these purposes. Real-world contact networks are rich in structural features that influence infection transmission, such as tightly-knit local communities that are weakly connected to one another. In this paper, we propose a new flow-based edge-betweenness centrality method for detecting bottleneck edges that connect nodes in contact networks. In particular, we utilize convex optimization formulations based on the idea of diffusion with p-norm network flow. Using simulation models of COVID-19 transmission through real network data at both individual and county levels, we demonstrate that targeting bottleneck edges identified by the proposed method reduces the number of infected cases by up to 10% more than state-of-the-art edge-betweenness methods. Furthermore, the proposed method is orders of magnitude faster than existing methods.

## Author summary

During the COVID-19 pandemic decision makers frequently face questions like where to impose a lockdown, which traffic to close, and whom to quarantine, all required to be carried out at minimal costs. Establishing cost-effective pandemic control policies requires identifying good targets. New computational models from network theory and epidemic simulations over real contact networks provide a valuable tool for finding the right bottlenecks to target upon. Here we study a computationally efficient network centrality measure that enables us to detect local transmission bottlenecks, i.e., contact edges that are especially important for the spread of disease among small communities or local network structures inside large networks. We find that pandemic intervention strategies that target at local network structures significantly outperform interventions that solely focus on the entire network structure as a whole, which are traditionally believed to be the most effective.

**Data Availability Statement:** All data files are available from figshare (DOI 10.6084/m9.figshare. 13166507) and/or GitHub (github.com/s-h-yang/ TargetedPandemicContainment). Computer code is

available from GitHub (github.com/s-h-yang/
TargetedPandemicContainment).

**Funding:** Research partially supported by a
Borealis AI Fellowship (https://www.borealisai.
com) to SY, and by NSERC-Discovery Grant
(RGPIN-2019-04067) to KF. The funders had no
role in study design, data collection and analysis,
decision to publish, or preparation of the
manuscript.

**Competing interests:** The authors have declared
that no competing interests exist.

## Introduction

Mathematical and computer simulation models of COVID-19 transmission are being widely
used during the COVID-19 pandemic for their ability to project future cases of infection
under various possible scenarios for mitigation strategies [1–4]. A significant subset of these
models are network simulation models [5–7]. In network models, the nodes of the network
represent individuals or population centres, and the edges represent contacts through which
SARS-CoV-2 (the virus that causes COVID-19) can spread. These models are often parameterized with data on demographic features, COVID-19 epidemiology, and population movement
patterns [8, 9]. Network models are particularly relevant to COVID-19 control through physical distancing measures. These measures are effective but socially and economically costly.
Therefore, physical distancing that targets the smallest number of nodes or edges of a contact
network required to achieve public health goals is desirable.

The dynamics of infection transmission on networks are known to be very different from
infection dynamics in homogeneously-mixing populations such as represented by compartmental epidemiological models [10–14]. For instance, network structure can change the epidemic threshold that determines whether the pathogen is able to spread across the network
[12] and spatial structure more generally can slow down the spread of the epidemic [15, 16].
Moreover, the contact structure of networks suggests control strategies that can exploit its
features. Previous models of infection control on networks have often concentrated on node-level characteristics such as node degree [17–20]. For instance, models can be used to explore
the impact of vaccination strategies that target highly connected nodes, or various different
approaches to contact tracing [17–20].

Earlier network modelling efforts focused on strategies for node-level characteristics
because data on the structure of entire contact networks was once rare. However, such data
is becoming increasingly available, making it possible to address strategies that target the
larger-scale features of network structure such as how connected communities are to one
another. It has been shown in simulated networks that vaccination targeted at individuals
that bridge different communities in the network are more effective than targeting individuals with high node degree [21]. These approaches detect important nodes and edges based on
edge-betweenness measures. In particular, edge-betweenness is a measure of the influence
an edge has over a diffusion process through the network (e.g., spread of infectious diseases).
A classical example is that of shortest-path (SP) betweenness, which quantifies edge importance based on the assumption that information spreads only along shortest paths. However,
it has been noted [21] that this approach can overlook important connections in a network.
For example, in Fig 1A we see that SP betweenness only recognizes the shorter "bridge" in
the middle, while it completely neglects the two slightly longer, but still highly influential,
side bridges.

Random-walk betweenness [22] fixes this problem of SP betweenness by assuming that
information spreads along random paths in the network while giving more weight to shorter
paths. It is also named current-flow (CF) betweenness [23] due to the relation to network electrical current flows. Fig 1B shows that edge-betweenness that takes into account all possible
walks captures the relative importance of all bridges. However, we note that social contact
activities in large networks tend to be local [24, 25], in the sense that a majority of individuals
are mostly active within only a few small communities formed by close relationships such as
families, friends and colleagues, etc. Thus, containing the spread of infectious diseases usually
requires identification and control of contact bottlenecks at a local scale (e.g., within various
small communities and their interconnections) rather than global. For example, cutting off all
three bridges in Fig 1B would be terribly ineffective at slowing down the disease spread in the

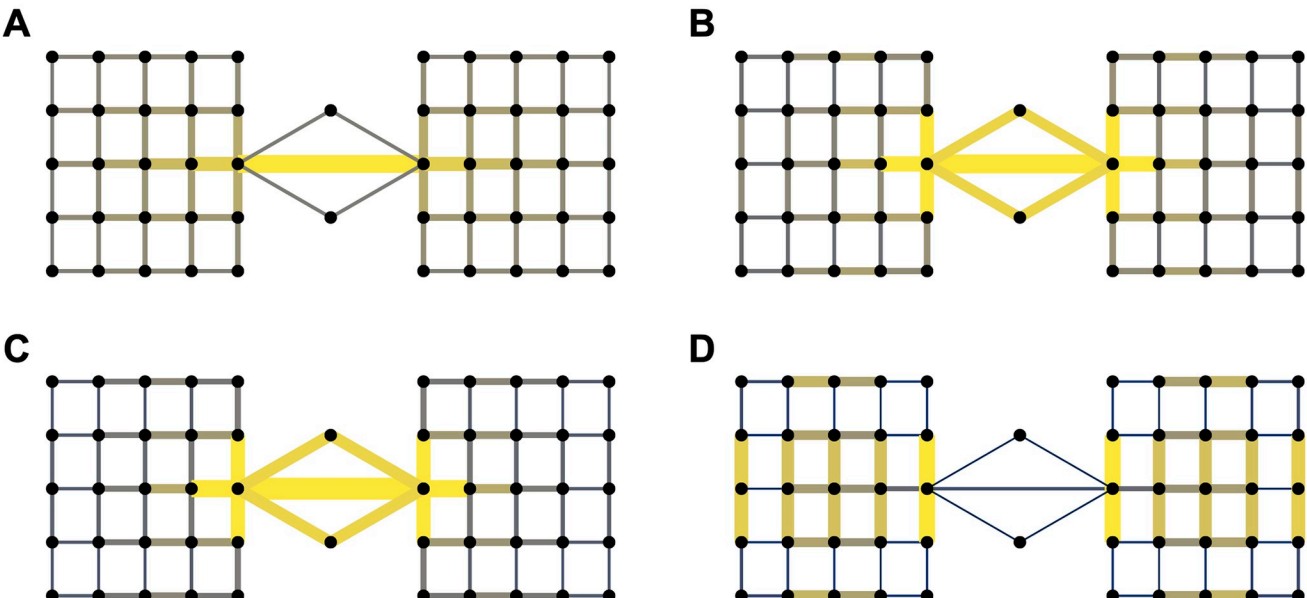

**Fig 1. Edge-betweenness measures.** Colour intensities and edge widths are chosen to reflect relative magnitude of betweenness measures. Yellow and thick edges have high betweenness scores, dark grey and thin edges have low betweenness scores. **(A)** The SP betweenness. It fails to identify the upper and lower "bridge" as important global bottleneck edges. **(B)** The CF betweenness. **(C)** The LF betweenness with λ = 1. It is almost identical to the CF betweenness. **(D)** The LF betweenness with λ = 2/5. It completely ignores the global bottleneck "bridges" and turns attention to edges within each block.

presence of community outbreaks, i.e., if there were already infectious nodes in each of the two "square" clusters.

From one point of view, removal of edges in a network corresponds most closely to non-pharmaceutical interventions (NPIs) that reduce contacts (edges) but do not change the nature of nodes. Examples of this include imposing travel restrictions between two cities, or a susceptible individual adopting contact precautions to prevent being exposed to an infected individual. In contrast, pharmaceutical interventions (PIs) such as vaccines and antiviral drugs change the nature of the nodes. Both NPIs and PIs are often employed to mitigate outbreaks of endemic infectious diseases, for which vaccines and drugs may already be available. But for a pandemic caused by a novel emerging pathogen, NPIs (i.e. edge removal, perhaps even including all edges emanating from a given node) are often the only means of combatting the pathogen until PIs become available. We focus on edge removal as a means to contain a pandemic caused by a novel emerging pathogens through NPIs, but we note that edge removal could be applied more broadly to any kind of epidemic containment.

In this paper we develop a new edge-betweenness measure for which we call it local-flow (LF) betweenness. It is based purely on local diffusion in the network and offers a very flexible and localized quantification of edge importance parametrized by λ ∈ (0, 1]. Intuitively, LF betweenness can be seen as a localized version of CF betweenness, where we assume that information spreads and also gradually fades away along random paths in the network. The parameter λ controls how fast information settles down and, hence, how far information can spread along edges in the network. When λ is large, it models the scenario where information can spread far away from any starting node; in this case, LF betweenness identifies global bottleneck edges in the entire network. When λ is small, it models the scenario where information quickly settles down near a starting node and thus cannot spread further away; in this case, LF betweenness tends to detect locally important edges as opposed to global bottlenecks that have little influence on local structure or transmission processes. Because of this, we will refer to λ

as the locality parameter. As a concrete example, Fig 1C shows that when λ equals 1, it detects the same global bottlenecks as identified by CF betweenness, but when we shrink λ to 2/5, it detects locally important bottlenecks within each block as shown in Fig 1D. Removing these bottlenecks would reduce disease transmission even if the infection is initially present in both sides. The proposed definition of edge-betweenness is based on $p$-norm flow diffusion [26]. This diffusion is defined as a convex optimization problem that models the phenomenon of diffusing mass from a given node to nearby nodes that have non-zero capacities. The origin of $p$-norm flow diffusion is in local graph clustering methods. Because of this, the proposed edge-betweenness method induces locality and clustering biases crucial to the good performance of effective pandemic containment.

We demonstrate that LF betweenness gives rise to better intervention strategies on three real datasets that we tested, and we discuss in detail why it is a more suitable measure for identifying disease transmission bottlenecks. We conduct exhaustive simulations and the conclusions we draw from all experiments are consistent.

## Results

We compare the effectiveness of interventions for the control of COVID-19 transmission that target edges meeting certain criteria. Specifically, we compare the following edge selection techniques: 1) Uniform (UI) intervention: target all contact edges uniformly, 2) High Degree (HD) intervention: target the contact edges incident to nodes having high degrees; 3) Eigenvector (EG) intervention: target the contact edges incident to nodes having high eigenvector centralities [27, 28]; 4) SP intervention: target the contact edges having high SP betweenness, 5) CF intervention: target the contact edges having high CF betweenness, and 6) LF intervention: target the contact edges having high LF betweenness. We use UI as a trivial baseline measure, and we consider HD and EG as two mildly nontrivial baseline measures. Node interventions based on the degree and eigenvector centralities have been studied in the context of network immunization [29]. However, because neither HD nor EG naturally applies to quantify edge importance, our simulation studies reveal that HD and EG are not suitable for edge interventions that we consider in this work. On the other hand, SP and CF betweenness offer intuitive quantifications of edge importance, and have been applied to a wide range of problems including cancer diagnosis [30], immunization modelling [21], power grid contingency analysis [31], terrorist networks analysis [32]. Hence, they are the best candidates for comparison purposes. Recall that our LF betweenness comes with a locality parameter $\lambda \in (0, 1]$, and different choices for λ lead to different quantifications of edge importance. In order to examine the effect of λ on the intervention results, we consider $\lambda \in \{1/2, 1/10, 1/50\}$ in our experiments. We will discuss how to pick λ at the end of this section.

Physical contact reduction is naturally modelled as edge weight reduction or edge deletion. Therefore, once a set of target contact edges is identified, we reduce the corresponding edge weights by 90%. We keep 10% weights on targeted edges in order to reflect some practical constraints, e.g., a minimal level of interaction may be required in case of emergency. In S6 Fig we show that reducing targeted edge weights by 99% produces similar results. We use two *Susceptible-Exposed-Infectious-Removed* (SEIR) network models to predict how COVID-19 infections will spread: 1) an ordinary differential equation (ODE) model where each node corresponds to a population in which an SEIR epidemic is occurring that can spread between nodes according to the network's adjacency matrix, 2) an agent-based model where each node corresponds to a person, and the infection is transmitted from one node to the next with a certain probability per time-step. For the population-based model, the interventions could represent selective road closure, travel screening, or quarantining towns and cities, as happened

during the Wuhan COVID-19 outbreak for instance. For the individual-based model, the interventions could represent public health measures that advise or incentivize individuals who are connected to a bottleneck to practice physical distancing.

We present the most informative discussions and figures in this section. We refer to S4–S6 Figs for additional experimental settings and simulation results that further support the effectiveness and robustness of the proposed edge intervention method based on LF betweenness. We refer to S2 Text and S8 Fig for an experiment using synthetic networks that further demonstrate why LF provides the most effective intervention targets. For the completeness of this study we also consider experiments for node immunization, as proposed in priori work on network epidemic interventions on nodes [29, 31]. We demonstrate that node interventions based on LF betweenness defined on nodes can outperform other competitive methods as well.

## Datasets

**Facebook county network [33, 34].**   This Facebook social network consists of 3,142 counties (nodes) and 22,138 edges. Two counties are connected with an edge if there exists strong social interaction between them as measured by Facebook interactions. In this report, out of all the edges we keep only those that correspond to counties less than 500 miles apart. We also removed geographically isolated states Hawaii and Alaska. The resulting graph is still a connected graph. The post-processed graph maintains the structural properties that are discussed in the original article and paper [33, 34], that is, social interaction tends to happen mostly among nearby counties. As a result, we treat the Facebook county network as a proxy for the frequency of physical contacts between individuals in different counties in the United States. However, we note that it cannot capture all aspects of physical cross-county interactions, such as those caused by long-distance commercial transport or those caused by individuals who do not use online social media, for instance.

**Wi-Fi hotspots Montreal network [35].**   This is a public Wi-Fi hotspot network that we interpret as a contact network, since each edge between two nodes in this network represents two hotspot users at the same location for some period of time. We note that it does not provide a representative sample of physical contacts in the general Montreal population. Wi-Fi networks are commonly used as proxies of human contact networks for studying transmission of infection across a network of individuals [35, 36] This particular network is by Île Sans Fil (ÎSF), a not-for-profit organization established in 2004 in Montreal, Canada, that operates a system of public internet hotspots. Each individual user is a node and concurrent usage of the same hotspot is an edge. We use the post-processed network by [35], which consists of 103,425 nodes and 630,893 edges.

**Portland, Oregon network [37].**   This synthetic network was generated from time use and census data for the city of Portland, Oregon. It has also been widely used in infectious disease modelling [20, 37, 38]. The full dataset consists of 1.6 million nodes and 31 million edges. Each individual person is a node and two persons are connected by an edge if they collocated at the same location during a short period of time. We also make use of a sub-sampled version of this dataset that has 10,000 nodes and 199,168 edges [20]. The reason that we sub-sample the original network is because the SP and CF betweenness methods do not scale to the initial network.

In Fig 2 we demonstrate the Network Community Profile (NCP), the degree distribution and epidemic curves without intervention. The NCP captures clustering pattern of a network, i.e., lower NCP means more significant clustering pattern in the network (see Methods for details). In Fig 2A we demonstrate that the datasets correspond to the three distinct NCP

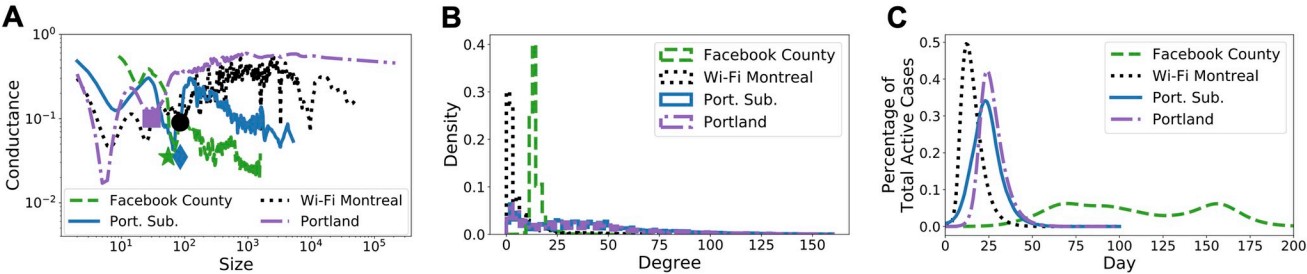

**Fig 2. Characteristics of the original networks. (A)** Network community profiles (NCPs, see Methods for a brief introduction). The NCPs have been computed using the *Local Graph Clustering* API [39, 40] based on the original paper [24]. The markers in the NCPs in Fig 2A correspond to tightly-knit clusters of nodes that we used to initialize the epidemic models for Fig 2C. **(B)** Degree distributions. **(C)** Predicted epidemic curves without intervention.

classifications from [24, 25]. In particular, Facebook County has a downward sloping NCP, i.e., conductance decreases as size increases, Wi-Fi Montreal has roughly flat NCP, i.e., conductance does not change much as a function of size, and Portland, Oregon has upward slopping NCP, i.e., conductance are small at small sizes and increases as the size increases. We will exploit the NCP structure to define the initially infected nodes in our experiments. In Fig 2B we illustrate the degree distribution for the datasets. Note that the degree distribution for Wi-Fi Montreal is heavily concentrated around nodes with degree $\leq 2$, which is more than half of the nodes in the network. This will play crucial role in the analysis of our experiments later on in this section. In Fig 2C we show the percentage of total active COVID-19 cases (prevalence of infection) against time (in days). The curve for Facebook County is very different from the other datasets because the data represent a nationwide geographic region and it takes a longer amount of time for the infection to spread from the Northeastern states to the rest of the country. This is also the reason that for Facebook County the curves have multiple peaks, since there are multiple outbreaks in multiple cities as the disease progresses. In contrast, the other datasets correspond to outbreaks in a single urban centre that tend to unfold over weeks instead of months.

## Experiments for Facebook County network

We apply the population-based ODE model to simulate the spread of COVID-19 on Facebook County network, since each node represents an entire county population. We assume that all county populations are initially susceptible and we pick infected counties for which we initialize 0.1% of the county population as infectious. We use three different ways to select initially infected counties to account for variations in where outbreaks could have started: (i) populated cosmopolitan cities, e.g., New York, Los Angeles, (ii) a tightly-knit cluster of 67 densely connected counties, highlighted in Fig 3A and also captured by the green star on the NCP in Fig 2A, and (iii) a random selection of 1% of all counties.

Simulation results under scenario (ii) is shown in Fig 3. Observe that the epidemic curves using LF intervention for $\lambda \in \{1/10, 1/50\}$ starts late and remains relatively flat, i.e., it has the lowest epidemic peak compared to any other intervention strategies. Also note that targeting edges according to the eigenvector centrality does not reduce the epidemic peak at all. Additional simulation results in S1 Fig also show that for all three different initial conditions and at all intervention coverage levels, LF method with a relatively small $\lambda \in \{1/10, 1/50\}$ leads to the most significant reduction in the epidemic sizes. We refer the reader to S4–S6 Figs for simulation results under different SEIR initial conditions, model parameters, edge weight reduction and intervention scenarios.

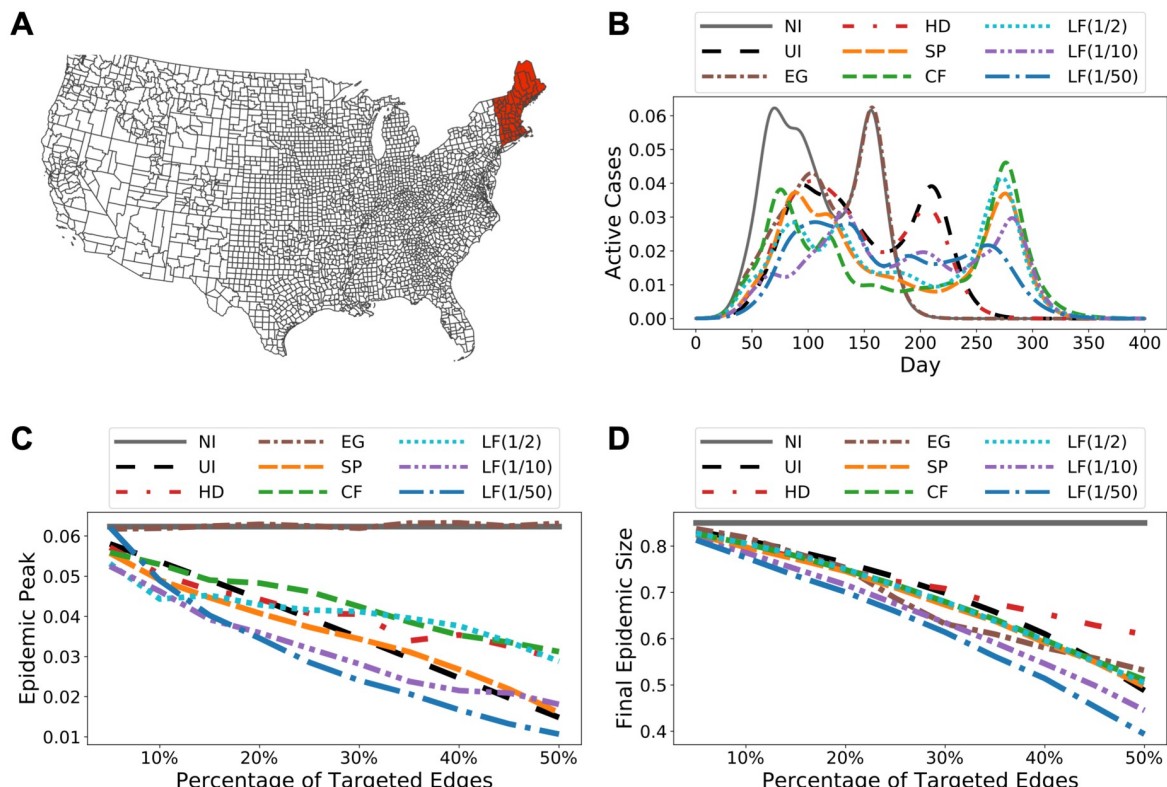

**Fig 3. Simulation results for Facebook County.** (**A**) Initially infected cluster of counties. (**B**) Predicted epidemic curves at 25% coverage level, i.e., intervention is applied to 25% of all edges, along with the original epidemic curve without intervention (NI). The abbreviations in the legend are introduced at the beginning of Results. LF(a), where $a \in \{1/2, 1/10, 1/50\}$, represents LF intervention with parameter $\lambda = a$. (**C**) Predicted epidemic peaks (peak prevalence) over a range of intervention coverage levels. (**D**) Predicted epidemic sizes (total infection) over a range of intervention coverage levels. Observe that EG intervention does not reduce peak prevalence at all. This is likely due to the epidemic curve has two modes and EG mostly targets on edges connected to the counties where the first peak happens, e.g., Fig 3B shows that EG significantly reduces the first epidemic peak but has almost no effect on the second. Fig 3A uses Plotly Python Open Source Graphing Library [41] with base map data from Natural Earth @ naturalearthdata.com. Direct link to base map data: https://www. naturalearthdata.com/http/www.naturalearthdata.com/download/110m/cultural/ne_110m_admin_1_states_provinces.zip.

To study what makes LF betweenness a much better indicator for local contact bottlenecks, we fix the coverage level at 25% of all edges and analyze the resulting networks. In Fig 4 we colour each county according to how many edges incident to it have been identified for contact reduction. We observe a significant difference in the patterns demonstrated by the three

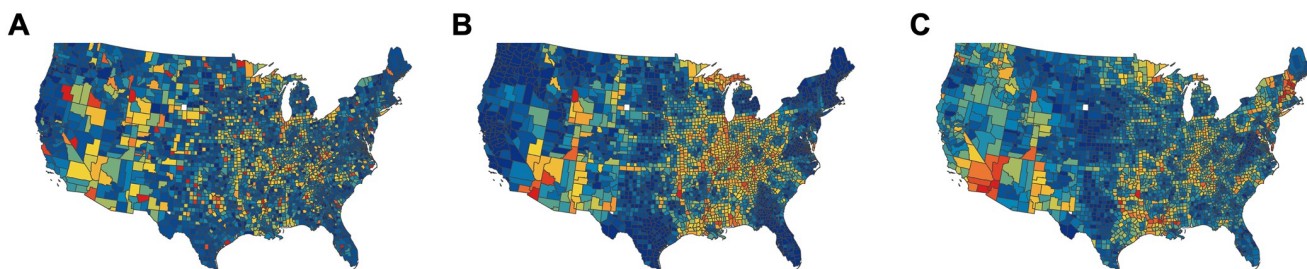

**Fig 4. Distribution of target edges reflected by county-level colours.** A county is coloured red if intervention is applied to most edges incident to it, a county is coloured dark blue if intervention is applied to very few edges incident to it. (**A**) SP intervention. (**B**) CF intervention. (**C**) LF intervention, $\lambda = 1/50$. The plots are made with base map data from Natural Earth @ naturalearthdata.com. Direct link to base map data: https://www.naturalearthdata.com/http/www.naturalearthdata.com/download/110m/cultural/ne_110m_admin_1_states_provinces.zip.

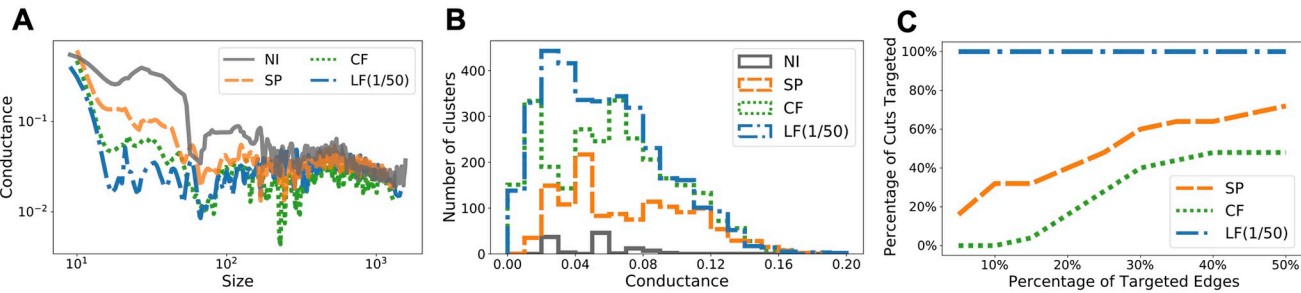

**Fig 5. Characteristics of the modified Facebook County networks due to targeted interventions using different edge-betweenness measures. (A)** NCPs (see Methods for a brief introduction). **(B)** Distribution of small-size clusters (having less than 100 nodes) by conductance. **(C)** Percentage of targeted out-link edges from the initial cluster of infection. Fig 5A and 5B show that the modified network due to LF intervention has many more well-defined (i.e., low conductance) small clusters, which can be effective at slowing down or stopping the spread of disease among tightly-knit small communities. For example, Fig 5C shows that only the top 5% edges based on LF betweenness already include all the out-link edges from the initial cluster of infection in Fig 3A.

methods. SP intervention results in scattered targets (i.e., counties coloured in red and orange) distributed over the entire country, whereas CF emphasizes the central east region which consists of a large number of concentrated small counties. Therefore, both methods demonstrate a global pattern as the targets of SP are dispersed over the entire network and the targets of CF are clustered in the middle. On the other hand, the targets of LF intervention constitute a few small groups of local clusters that spread across the country and loosely partition both east and west coasts into several smaller connected components. This observation is further supported in Fig 5A in which the NCP of the modified graph based on LF has much lower conductance when cluster sizes are small. In Fig 5B we investigate this range by plotting the distribution of clusters of size less than 100 against conductance. Not surprisingly, the resulting network based on LF intervention contains more well-defined small clusters than the networks obtained from SP or CF method, which has a more global focus. Finally, in Fig 5C we measure the percentage of out-link edges from the initial infected cluster (cf. Fig 3A) that are targeted by different intervention strategies. Observe that the top 5% edges based on LF betweenness already include all edges in the cut of the initial cluster. This explains why the epidemic curve under LF intervention starts rising later than others: because all out-link contacts have already been reduced. Note that LF intervention is un-supervised, i.e., the method is not aware of the initially infected nodes. This demonstrates that LF betweenness has a strong local clustering bias. We formalize this in Methods.

## Experiments for Wi-Fi Montreal network

We apply the agent-based SEIR network model to Wi-Fi Montreal, since each node now represents an individual person. We assign the initial state *Susceptible* to each person and then pick *Infectious* persons in two ways that cover very distinct scenarios: (i) as a group of 120 densely connected persons captured by the black circle on the NCP in Fig 2A, and (ii) as 0.1% of total population selected uniformly at random. We simulate the model until all state transitions reach equilibrium. The results for scenario (i) are shown in Fig 6 (see S2 Fig for similar results for scenario (ii)). Observe that in most cases, and in particular when targeting more than 20% of contact edges, LF intervention for $\lambda \in \{1/10, 1/50\}$ significantly reduces both epidemic size and epidemic peak.

CF intervention is omitted for this network due to prohibitive computation time for computing CF betweenness. We note that computing SP betweenness for Wi-Fi Montreal took more than four days, and computing CF betweenness would take $O(\log|V|)$ more time. As a comparison, computing LF betweenness for $\lambda = 1/50$ was done under 10 minutes. We now

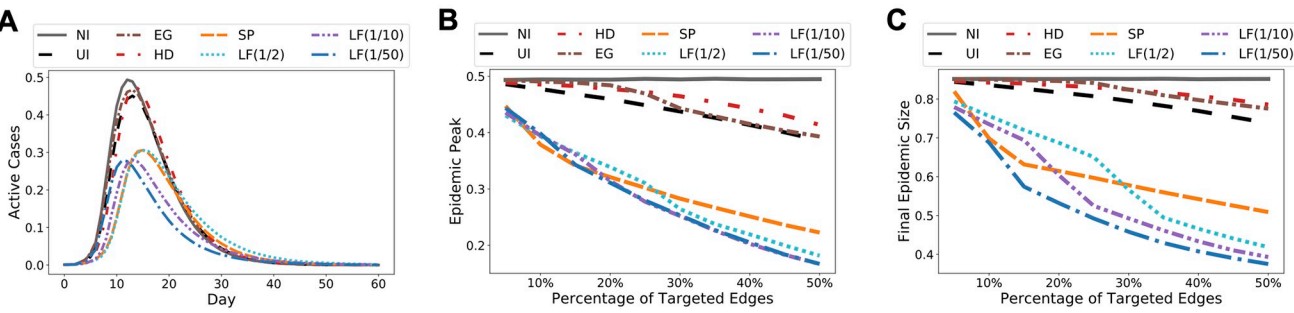

**Fig 6. Simulation results for Wi-Fi Montreal. (A)** Predicted epidemic curves at 25% coverage level. **(B)** Predicted epidemic peaks over a range of intervention coverage levels. **(C)** Predicted epidemic sizes over a range of intervention coverage levels.

explain qualitatively what makes LF intervention work better than SP intervention. As discussed earlier, more than half of the nodes in Wi-Fi Montreal have degree one or two (cf. Fig 2B), perhaps because these nodes represent tourists and visitors. Hence, this network presents an extreme case where disconnecting all those small degree nodes could be a trivial yet effective solution. On the other hand, partitioning the entire graph into groups of clusters may not be as effective as it is for Facebook County. In Fig 7 we demonstrate that LF betweenness captures the degree irregularity in Wi-Fi Montreal and exploits this local information (i.e., many nodes have low degree). In particular, Fig 7A shows that LF intervention does not necessarily generate more small clusters when the underlying graph has too many degree-one nodes. This is supported by Fig 7B where we see that the distribution of clusters of all sizes in the modified networks are similar. On the other hand, as shown in Fig 7C where we measure how many isolated singleton nodes are there if we were to remove all targeted edges, we notice that LF intervention separates far more singletons than SP intervention does, thanks to its locality bias (see Methods). The flexibility of incorporating local information (or going global if necessary, by controlling the value of λ) is what makes LF betweenness versatile and effective.

## Experiments for Portland, Oregon

We apply the agent-based model on both sub-sampled and full Portland contact networks. We consider sub-sampled network (Port. Sub.) because the computation of both SP and CF betweenness measures do not scale to the full Portland dataset. We use Port. Sub. for comparing different intervention methods and full Portland to demonstrate the effectiveness of LF

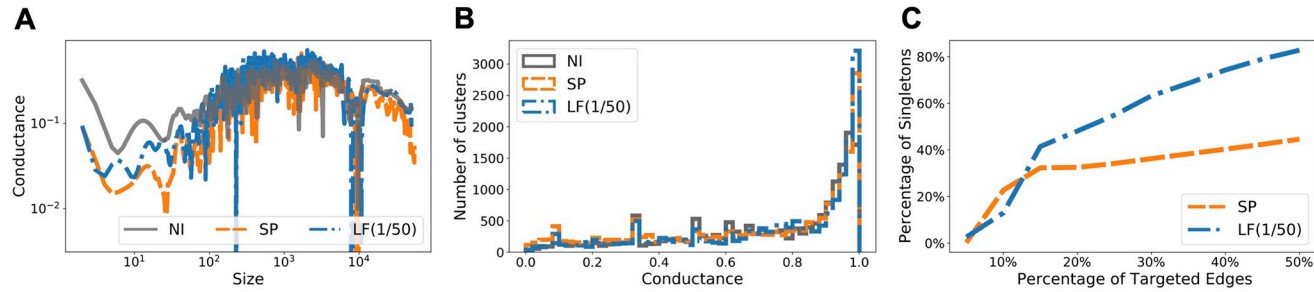

**Fig 7. Characteristics of the modified Wi-Fi Montreal networks due to targeted interventions using different edge-betweenness measures.** Locality bias of LF betweenness for Wi-Fi Montreal is illustrated by the dramatic difference in Fig 7C: When the network has too many singleton nodes, LF intervention isolates those nodes. **(A)** NCPs (see Methods for a brief introduction). **(B)** Distribution of clusters by conductance. **(C)** Isolation of singleton nodes.

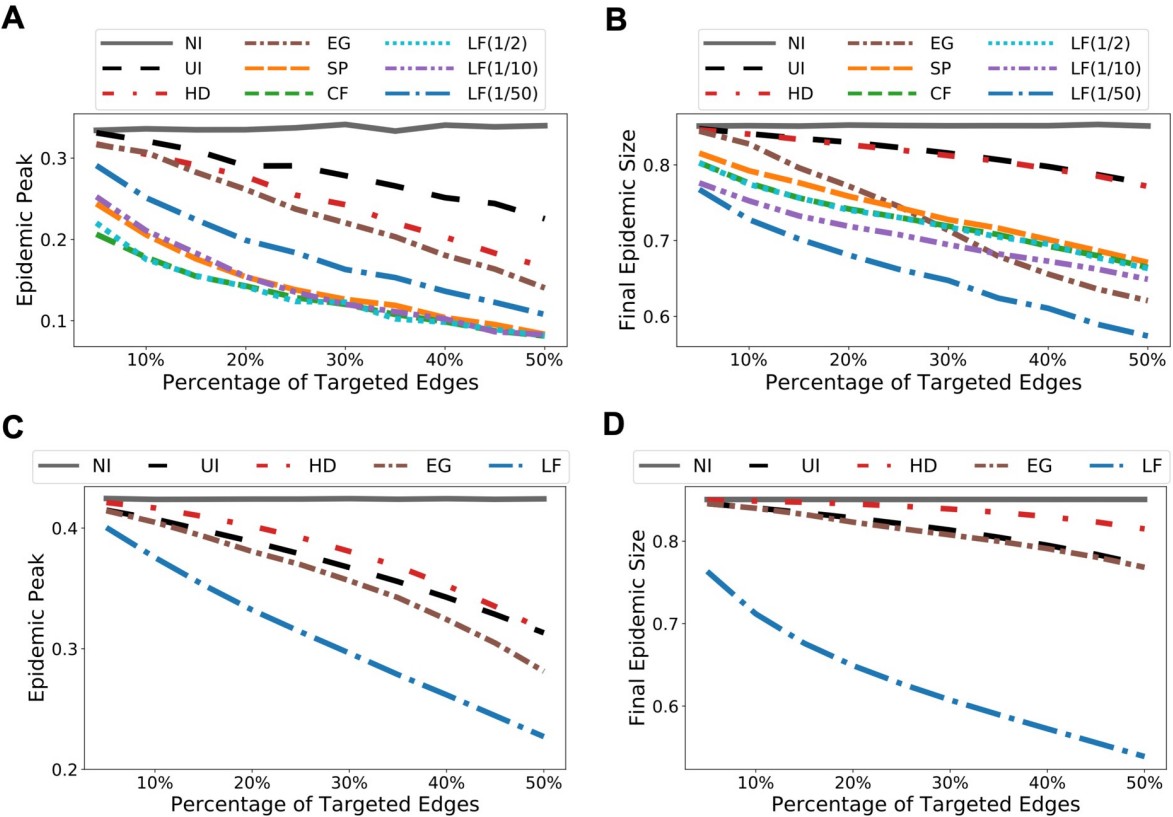

**Fig 8. Simulation results for both sub-sampled and full Portland networks.** For the full Portland network we used $\lambda = 1/1000$ for scalability. **(A)** Predicted epidemic peaks, sub-sampled network. **(B)** Predicted epidemic sizes, sub-sampled network. **(C)** Predicted epidemic peaks, full network. **(D)** Predicted epidemic sizes, full network.

method after scaling it up for large networks. We consider two initialization techniques for the model. First, we use well-connected clusters illustrated by the purple square and the blue diamond on the NCP in Fig 2A. Second, we select randomly 0.1% nodes from the entire population. For both datasets, the simulation results for cluster initialization are shown in Fig 8. The results obtained from random initializations are almost identical and we leave them to S3 Fig. Observe that the smaller the $\lambda$, the smaller the total epidemic size. On the other hand, there is a trade-off between epidemic peak and epidemic size: For Port. Sub., $\lambda = 1/50$ gives the most reduction in epidemic size, whereas a slightly larger $\lambda = 1/10$ offers less reduction in total infection but gives a flatter epidemic curve (i.e. lower peak).

For Port. Sub., all three methods produce similar NCPs when cluster sizes are more than 30 as shown in Fig 9A. So NCP does not explain why LF intervention leads to the most reduction in epidemic size. We further investigate the degree distributions in Fig 9B. Notice that, after LF intervention, more than 30% of all nodes have degrees close to 0, which is more than double the amount created from SP or CF method. This large amount of *almost-isolated* nodes (as they have degrees close to 0) makes it very difficult for an epidemic to spread across the entire population, and explains why LF intervention leads to the mildest outbreak in terms of total infection. It also reveals that LF betweenness offers a better utilization of "budget" in the sense that most efforts in contact reduction are spent to create and isolate low degree nodes. Finally, for the full Portland network, while Fig 9C shows that there is a small difference in NCP, such difference is not as significant as it is demonstrated on Facebook County network, and the

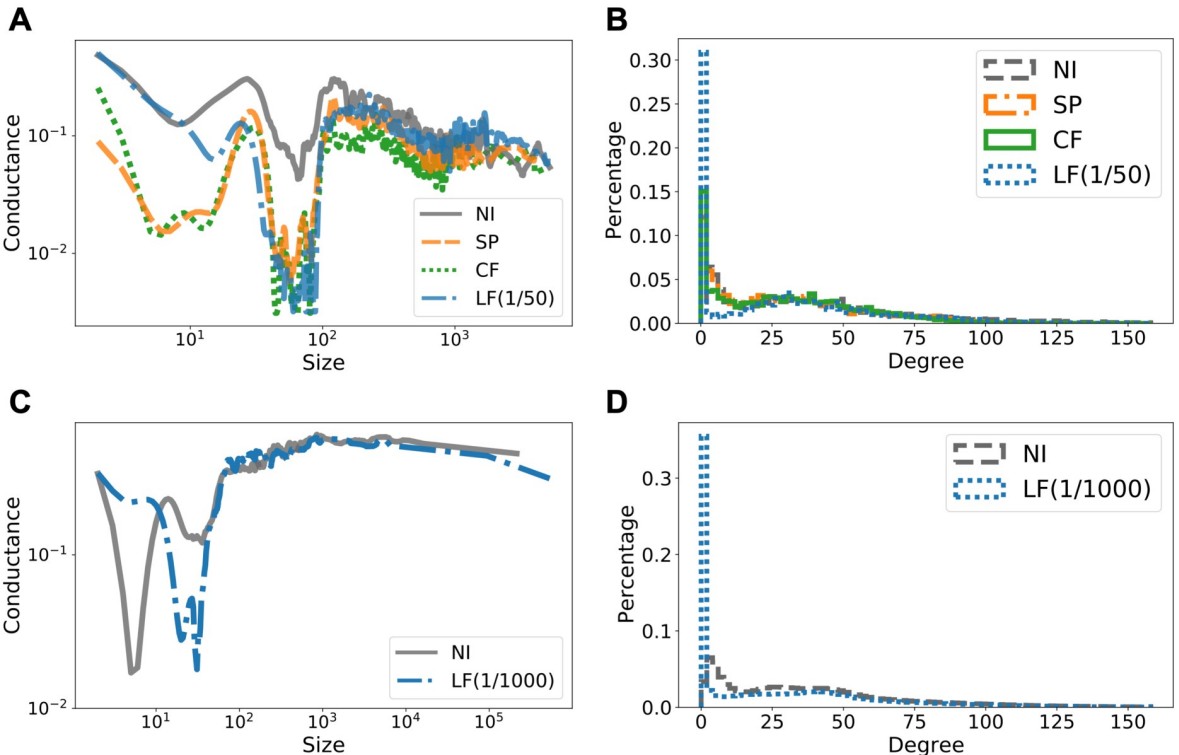

**Fig 9. Characteristics of the modified Portland networks due to targetedinterventions.** LF intervention causes the networks to contain significantly more nodes having low degrees, and thus greatly reducing the probability of disease spread over these nodes. **(A)** NCPs, sub-sampled network. **(B)** Degree distributions, sub-sampled network. **(C)** NCPs, full network. **(D)** Degree distributions, full network.

major benefit of using LF intervention on the full network still lies in the large amount of low degree nodes it created, as we show in Fig 9D.

## Robustness of LF intervention under different model or intervention settings

We will provide details on model parametrization in Methods, but let us now discuss the robustness of our simulation results. First of all, both population-based and agent-based models have been parametrized so that 85% of the population would be affected without intervention. This choice of final epidemic size is based on parametrizing the transmission rate $\beta$ to achieve a basic reproduction rate $R_0 = 2.5$ (see Methods) on the Portland contact network. In order to examine the effectiveness of LF intervention in scenarios where the pandemic has a lower final infection size, we carried out additional experiments where $\beta$ is parametrized so that the final sizes are 70% and 55%, respectively. Fig 10 shows the simulation results for each dataset when the epidemic size is 55% of the population without intervention. Observe that in this case, for most intervention coverage levels, LF betweenness still outperforms other network measures. UI, HD, and CF occasionally produce better results, but their overall performances are not consistent. The results in S4 Fig for model parameterizations that reach 70% final size without intervention are similar.

Besides robustness against variations in model parameterizations, one may be interested in scenarios where the intervention methods are not implemented from the start of a pandemic. This could be the case during a pandemic with multiple waves of outbreak and as a result

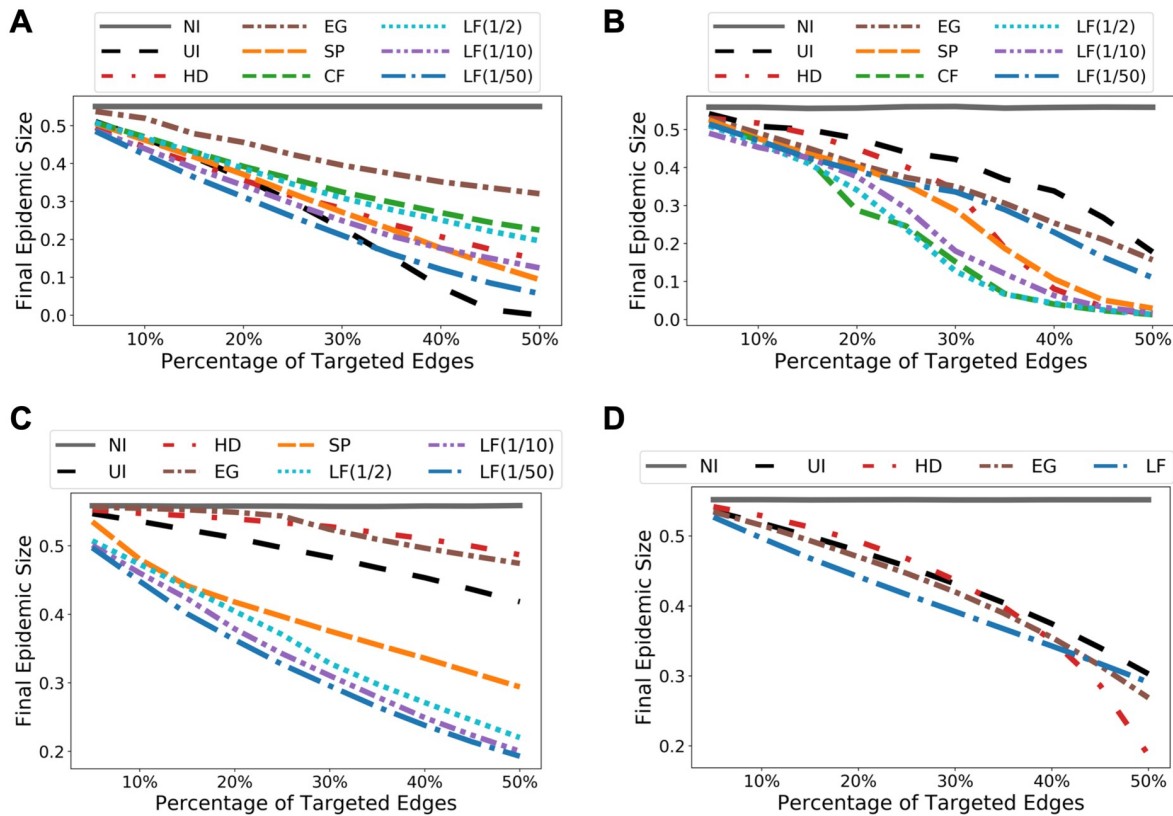

**Fig 10. Simulation results under alternative model parametrization.** The transmission rate parameter is calibrated so that 55% of the population would be affected without intervention. We initialized both population-based and individual-based models using random initialization, where a few randomly selected nodes are labelled as infectious at time 0. The plots show for each dataset the final epidemic sizes under different intervention strategies and various percentage coverages. We average over 50 runs for random initialization. **(A)** Results for Facebook County. LF for $\lambda \in \{1/10, 1/50\}$ still leads to the most reduction in epidemic sizes when the coverage level is less than 30%. When targeting more than 30% edges, uniform intervention (UI), which uniformly reduces the transmission rate over all edges, becomes more effective for Facebook County. This is because the transmission rate parameter is getting close to a threshold under which a pandemic would not emerge, which is seen in the 0% final size under UI at 50% coverage level. **(B)** Results for Wi-Fi Montreal. LF interventions still lead to the most reduction in the final sizes. **(C)** Results for Port. Sub. CF and LF with $\lambda = 1/2$ have the best performance overall, while LF with $\lambda = 1/10$ is the most effective when targeting less than 15% edges. **(D)** Results for Portland. We used $\lambda = 1/1000$ in order to scalably compute LF. We see that LF has better performance when targeting no more than 40% of all edges.

public health policies will have to change accordingly. In order to study the effectiveness of intervention methods when they are applied in the middle of an outbreak when one already observes an exponential growth in the number of infections, we conducted additional experiments and we refer the reader to S5 Fig for a complete set of results. In this case, LF intervention is still the most effective.

Finally, the results we have shown so far are based on reducing the transmission rate (edge weight) on targeted edges by 90%. In practice, one may impose looser or stricter reductions depending on the level of intervention coverages. For example, when targeting a small portion of highly important bottlenecks, 90% reduction in contact rate may not be strict enough for effective pandemic mitigation. In this regard, we conducted additional experiments where the targeted edge weights are reduced by 99%. We refer the reader to S6 Fig for a complete set of results which show that LF intervention still delivers the best overall performance.

## Experiments for node immunization

Since all the centrality and betweenness based edge selection methods we considered so far also apply to quantify node importance, we demonstrate that, in the context of node immunization where a small set of selected nodes are immunized, node selection according to LF betweenness also delivers the best performance overall. We compare LF with the following methods: selecting a set of nodes uniformly at random (UI), targeting nodes having high degree centralities (HD), eigenvector centralities (EG), shortest-path betweenness (SP), and current-flow betweenness (CF), respectively. Once a set of nodes is selected, we "immunize" a node by disconnecting it from the rest of the network. Simulation results (cf. Fig 11) show that the performance of LF matches CF on Port. Sub. network and outperforms all other methods. Again, the computation time for LF is orders of magnitude faster than CF, making it the only betweenness measure that scales to the full Portland network. Similar results on Facebook County network, where node "immunization" may represent a complete lockdown of a county/city, and Wi-Fi Montreal network, are shown in S7 Fig.

## Discussion

The comprehensive experiments we conducted in this section show that intervention strategies that rely on LF betweenness are more effective than interventions based on other network centrality measures. We believe that LF should be considered as a better identifier for epidemic transmission bottlenecks than other measures such as SP, CF betweenness, degree and

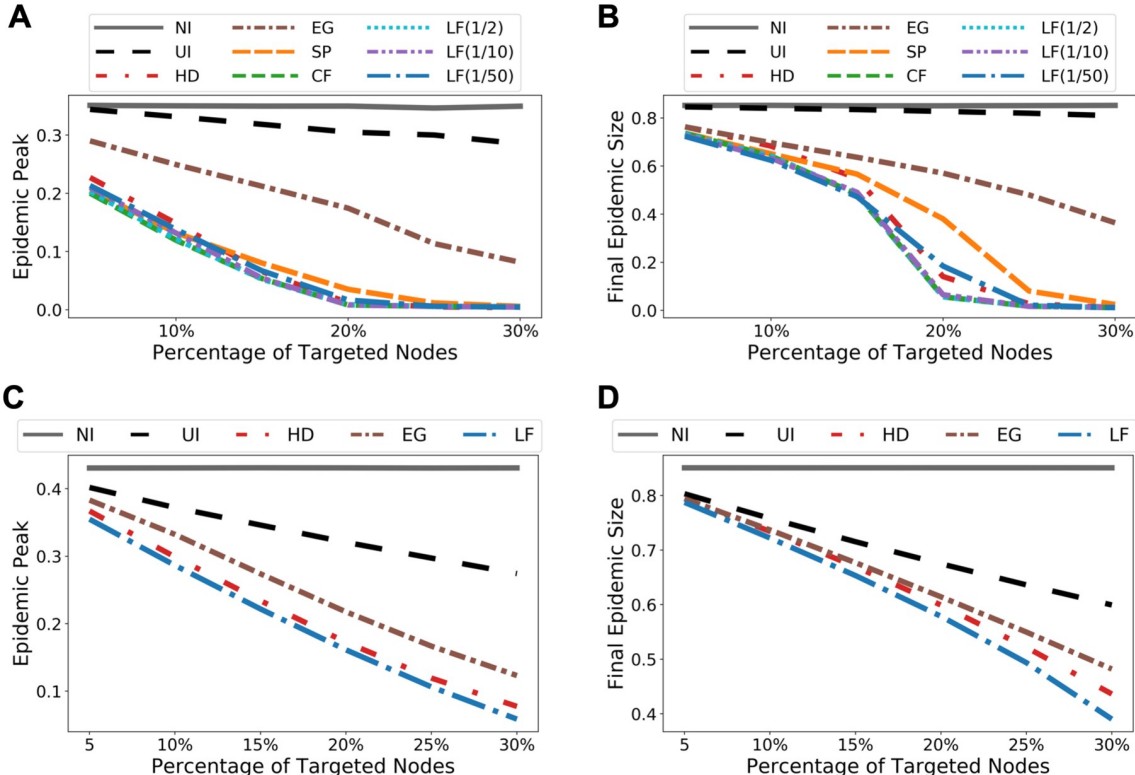

**Fig 11. Simulation results for node immunization on sub-sampled and full Portland networks.** The results are obtained from agent-based SEIR model with random initializations and are averaged over 50 trials. **(A)** Predicted epidemic peaks, sub-sampled network. **(B)** Predicted epidemic sizes, sub-sampled network. **(C)** Predicted epidemic peaks, full network. **(D)** Predicted epidemic sizes, full network.

eigenvector centralities, which have already been extensively exploited in priori work on network intervention strategies [21, 29, 42, 43]. In the context of pandemic mitigation, on the one hand, LF betweenness can be straightforwardly used to identify good targets for static intervention strategies similar to what we considered in this section; on the other hand, it can be incorporated into more complex and dynamic intervention methods, for example, sequentially remove nodes or edges similar to [43], or continuously adjust the percentage of edge weight reduction depending on the resulting LF betweenness measure of weighted networks. Exploiting LF betweenness and developing more sophisticated dynamic intervention strategies that uses LF for more effective pandemic mitigation methods would be an interesting future work.

Finally, let us discuss how to set $\lambda$ for LF betweenness. For epidemics that result in high final outbreak sizes without the presence of intervention, e.g., the COVID-19 pandemic, our experiments indicate that a small $\lambda$, e.g., $\lambda = 1/50$, often leads to the most significant reduction in outbreak sizes. Intuitively, the smaller $\lambda$ is, the more localized the corresponding LF betweenness is. As we see in the experiments, the ability to detect locally important bottleneck edges is an important contributor for the effectiveness of LF intervention. Recent study has also shown that local-scale intervention strategies outperform global-scale intervention strategies during the COVID-19 pandemic [1]. Therefore, a very crude way is to pick a reasonably small $\lambda$ like 1/50 or 1/10 that we have used in our experiments, because smaller $\lambda$ induces stronger locality bias in the LF betweenness, and thus the targeted edges are more local. On the other hand, as we see in Fig 10C, in some settings the results are sensitive to specific choice of $\lambda$, and we may require a larger $\lambda = 1/2$ to get the best overall intervention performance. For example, on the individual-based Port. Sub. network, for an epidemic parameter setting that gives lower final outbreak size without the presence of intervention, $\lambda = 1/2$ works better than smaller $\lambda$'s overall. In general, the 'best' $\lambda$ can depend on the nature of a network (e.g., population-based or individual-based), specific datasets, and estimated epidemic model parameters. Therefore, in order to select a good $\lambda$ value that leads to effective intervention over real-world networks, in practice one may try a number of different $\lambda$ values or perform grid search over an interval: Simply pick the $\lambda$ value that gives the best simulated intervention performance using the original or sub-sampled networks.

## Methods

### Baseline network edge-betweenness measures

Network edge-betweenness can be regarded as a measure of the extent to which an edge has control over the information that are passed through it. The simplest and one of the most widely used edge-betweenness measure is the shortest-path betweenness. Consider an undirected graph $G = (V, E)$ where $V$ is the set of nodes and $E$ is the set of edges. For two arbitrary nodes $s, t \in V$, let $\sigma_{st}$ denote the total number of shortest paths between $s$ and $t$; further, for $e \in E$, let $\sigma_{st}(e)$ denote the number of shortest paths between $s$ and $t$ that pass through $e$. Then the SP betweenness for $e$ is given by

$$\mathbf{bet}^{\mathrm{SP}}(e) := \sum_{s,t \in V, s \neq t} \frac{\sigma_{st}(e)}{\sigma_{st}},$$

where $n = |V|$ is the number of nodes. While SP betweenness is intuitive and simple, in most networks however, information (or disease) does not spread only along geodesic paths. The current-flow betweenness [22, 23] was introduced to model the phenomenon that information spreads along random paths in the network. Formally, [23] defines the CF betweenness using electrical currents over networks. Let $\tau_{st}$ denote the electrical $st$-current that stems from a unit

source $s \in V$ and a unit sink $t \in V$, and hence the quantity $|\tau_{st}(e)|$ corresponds to the fraction of a unit $st$-current flowing through $e$. The CF betweenness for an edge $e \in E$ is given by

$$\mathbf{bet}^{\mathrm{CF}}(e) := \sum_{s,t \in V, s \neq t} |\tau_{st}(e)|.$$

In our experiments we used two simple baseline centrality measures that typically apply to nodes. The first one is the degree centrality, which quantifies node importance according to node degrees. The second one is eigenvector centrality, which quantifies node importance according to the entries in the eigenvector corresponding to the largest eigenvalue of the adjacency matrix of the graph. In order to adapt the degree and eigenvector information to quantify edge importance, we define the corresponding edge score for $e = (u, v)$ by taking the maximum of incident node scores:

$$\mathbf{bet}^{\mathrm{HD}}(e) := \max\{d_u, d_v\}, \quad \mathbf{bet}^{\mathrm{EG}}(e) := \max\{x_u, x_v\},$$

where $d_v$ is the degree of node $v$ and $x_v$ is the entry in the eigenvector $x$ that corresponds to node $v$.

## Local-flow betweenness

In this section we formally introduce LF betweenness and discuss its locality and clustering biases. LF betweenness builds on $p$-norm flow diffusion [26], which originates as a tool to solve the local graph clustering problem [44] where the goal is to detect small clusters around a given set of nodes. There exist spectral [45–49] and combinatorial [44, 50–53] methods for local graph clustering. Spectral methods in general are computationally more efficient but have inferior clustering guarantees than combinatorial methods; combinatorial models usually require intricate tuning of parameters and thus are not suitable for a generalization to network betweenness measures. On the other hand, $p$-norm flow diffusion is as simple and as fast as spectral methods, while having better clustering guarantees both in theory and in practice. For these reasons, we use it to define our edge-betweenness measure. Moreover, as we will see later in Remark 1, the proposed general definition of edge-betweenness subsumes CF betweenness as a special case.

Given an undirected graph $G = (V, E)$ where $V$ is the set of nodes and $E$ is the set of edges. We are interested in the following diffusion process on $G$, which is formulated as a convex optimization problem [26]:

$$
\begin{aligned}
\text{minimize} \quad & \sum_{(u,v) \in E} f(u, v)^2 \\
\text{subject to} \quad & \sum_{v \in V:(u,v) \in E} f(u, v) + \Delta(u) \leq T(u), \quad \forall u \in V.
\end{aligned}
\tag{1}
$$

Intuitively, the optimization problem (1) models the process of spreading a given initial mass from some nodes to nearby nodes along the edges in the graph. Here, $\Delta$ and $T$ are vectors of length $|V|$ and they specify the amount of initial mass and sink capacity at each node, respectively. For example, $\Delta(u)$ and $T(u)$ denote the amount of initial mass and sink capacity at node $u$, respectively. The vector $f$ are flow variables of length $|E|$. For each edge $e = (u, v) \in E$, the corresponding entry $f(u, v)$ specifies the amount of mass that flows over $e$, and the sign indicates whether the mass flows in the forward or reverse direction of the edge $e = (u, v)$, i.e., $f(u, v)$ is positive if mass flows from $u$ to $v$ and vice versa. We abuse the notation to also use $f(v, u) = -f(u, v)$ for an edge $e = (u, v)$. Therefore, the quantity $\sum_{v \in V:(u,v) \in E} f(u, v) + \Delta(u)$ gives the

amount of final mass at node $u$ if we start with $\Delta(u)$ amount of initial mass at node $u$ and distribute the mass around according to flow routing $f$. We call a flow $f$ feasible if the final mass at each node is at most its sink capacity. The objective of problem (1) is to find a feasible flow that also has the minimum $\ell_2$-norm, which will be denoted by $f^*_{\Delta,T}$. We use subscript $\Delta$ and $T$ to emphasize its dependence on $\Delta$ and $T$. Naturally, in a diffusion process we start with $\Delta$ having high density, i.e., there is a large amount of initial mass concentrated on a small set of nodes, and the sink capacities enforce we spread the mass to get lower density.

The formulation described in problem (1) is the $p$-norm flow diffusion [26] when $p = 2$. We use $p = 2$ because it has fast computation and good empirical performance (see Results). Now we discuss how to exploit problem (1) and define a proper betweenness measure. We will start by defining a more general class of betweenness measures and then obtain both CF and LF betweenness measures as special cases. To take into account all relevant diffusion processes that start from arbitrary nodes and arbitrary sink capacities, we consider $\Delta$ and $T$ in problem (1) as random variables following a joint probability distribution $\mathcal{P}$, under which its expected optimal objective value is finite. We define, in the most general sense, the $\ell_2$-*flow edge-betweenness* for an edge $e$ as

$$\mathbf{bet}^{\ell_2-\text{flow}}(e;\mathcal{P}) := \mathbb{E}_{(\Delta,T)\sim\mathcal{P}}[|f^*_{\Delta,T}(e)|], \tag{2}$$

where we use $|f(e)|$ to denote the magnitude of flow over an edge $e = (u, v)$, i.e., $|f(e)| = |f(u, v)| = |f(v, u)|$. Of course, the specific inductive biases of $\ell_2$-norm flow edge-betweenness depend on the distribution $\mathcal{P}$. For example, let $\mathbf{1}_v$ denote the indicator vector of $v \in V$, i.e., $[\mathbf{1}_v]_u = 1$ if $u = v$ and 0 otherwise, and let $\mathcal{U}_V$ denote the discrete uniform distribution on the set of indicator vectors $\{\mathbf{1}_v : v \in V\}$, then one obtains the CF betweenness as a special case (see S1 Text for a formal argument):

**Remark 1**. *For an edge $e \in E$, the CF betweenness* [23] $\mathbf{bet}^{\text{CF}}(e)$ *normalized by* $1/|V|^2$ *satisfies*

$$\mathbf{bet}^{\text{CF}}(e) = \mathbf{bet}^{\ell_2-\text{flow}}(e;\mathcal{U}_V \times \mathcal{U}_V) = \mathbb{E}_{\Delta\sim\mathcal{U}_V, T\sim\mathcal{U}_V}[|f^*_{\Delta,T}(e)|].$$

In order to introduce locality and clustering bias in Eq (2), we consider the initial source vector as randomly drawn from the distribution $\mathcal{U}_V$, and we fix $T = \frac{d}{\lambda\cdot\text{vol}(G)}$ where $d$ is the degree vector, vol($G$) equals the sum of degrees of all nodes in $G$, and $\lambda \in (0, 1]$. We call the resulting specialized $\ell_2$-norm flow edge-betweenness as *local-flow betweenness* with parameter $\lambda$. More explicitly, for $s \in V$, let $f^*_s$ denote the optimal solution of problem (1) when we fix $\Delta = \mathbf{1}_s$ and $T = \frac{d}{\lambda\cdot\text{vol}(G)}$, then the LF betweenness for an edge $e \in E$ is given as

$$\mathbf{bet}^{\text{LF}}(e) := \mathbb{E}_{\Delta\sim\mathcal{U}_V}\left[|f^*_{\Delta,T}(e)| \;\middle|\; T = \frac{d}{\lambda\cdot\text{vol}(G)}\right] = \frac{1}{|V|}\sum_{s\in V}|f^*_s(e)|.$$

Intuitively, the LF betweenness of an edge $e$ is the expected amount of mass that would flow over $e$ if we diffuse a unit amount of initial mass from a randomly chosen node $s \in V$ to the rest of the graph. The magnitude of $\lambda \in (0, 1]$ in the sink capacities $T = \frac{d}{\lambda\cdot\text{vol}(G)}$ determines how far away the initial mass at $s$ can spread. More precisely, we make the following remark that the locality of edge flows is controlled by $\lambda$.

**Remark 2** (adapted to our problem from Fountoulakis *et al.* [26]). *Consider the optimal flow routing $f^*_s$ for any $s \in V$. We have that the number of edges with nonzero amount of mass routed over them is bounded by* $|\{e \in E : f^*_s(e) \neq 0\}| < 2\lambda|E|$.

We provide further interpretation for Remark 2. For $u \in V$, recall that $\Delta(u)$ specifies the amount of initial mass at node $u$. Therefore, for a fixed node $s$, the setting $\Delta = \mathbf{1}_s$ means that, initially, there is exactly one unit amount of mass at node $s$ and zero mass at other nodes. The

corresponding optimal flow $f_s^*$ specifies how the initial mass at $s$ are diffused to the rest of the graph. In this sense, Remark 2 says that $\lambda$ controls how far away from $s$ the initial mass can be sent to. When $\lambda$ is small, only a small number of edges will have nonzero flow crossing them, therefore the initial mass cannot spread too far away from $s$. On the other hand, if $\lambda$ is large, then many more edges will have nonzero flow crossing them, which implies that the initial mass are routed to larger regions in the graph as opposed to staying close to $s$. We refer the reader to S9 Fig for a concrete example on how $\lambda$ controls the locality of individual edge flows.

Besides locality, one can show that $f_s^*$ induces a local graph clustering bias for appropriately chosen $\lambda$. This local graph clustering bias plays a crucial role in our experiment. Formally, we quantify how "well-knit" a cluster is by measuring its conductance. The conductance of a subset of nodes $S \subseteq V$ is defined as

$$\phi(S) := \frac{|\partial(S)|}{\min\{d(S), d(V \setminus S)\}}, \qquad (3)$$

where $\partial(S) = \{(u, v) \in E: u \in S, v \notin S\}$ and $d(S) = \sum_{v \in S} d_v$ is the sum of node degrees in $S$. We state the following Remark 3 which connects Eq (1) with local clustering in terms of conductance. Because of the close relationship between primal and dual optimal solutions in general, an intuitive way to interpret Remark 3 is that the local clustering structures are encoded in $f_s^*$.

**Remark 3** (adapted to our problem from Fountoulakis *et al.* [26]). *Fix $T = \frac{d}{\lambda \cdot \mathrm{vol}(G)}$, and $\Delta = \mathbf{1}_s$ for some node $s$. The optimal solution to the dual of problem (1) gives a cluster $\tilde{C}$ such that the conductance $\phi(\tilde{C}) \leq \mathcal{O}(\alpha \cdot \sqrt{\phi(C)})$ holds simultaneously for any subset $C$ containing $s$, where $\alpha = \mathcal{O}\left(\frac{\lambda \mathrm{vol}(G)}{d_s}\right)$ and $d_s$ is the degree of $s$. In particular, when we set $\lambda = \mathcal{O}\left(\frac{1}{\mathrm{vol}(G)}\right)$, the guarantee becomes $\phi(\tilde{C}) \leq \mathcal{O}(\sqrt{\phi(C)})$.*

### Efficient computation of LF betweenness

Given $\Delta$ and $T$, an $\epsilon$-accurate solution to the dual problem of Eq (1) can be computed in time $\mathcal{O}\left(\lambda |E| \bar{d}^2 \log \frac{1}{\epsilon}\right)$ where $\bar{d}$ is an integer that satisfies $\bar{d} \leq \max_{i \in V} d_i$ [26]. The optimal solution $f_{\Delta,T}^*$ can be obtained in a straightforward manner from the optimal dual solution as follows. Let $x_{\Delta,T}^*$ be an optimal solution to the dual problem of Eq (1) [26]:

$$\text{minimize } \frac{1}{2}x^T L x + (\Delta - T)^T x \text{ subject to } x \geq 0, \qquad (4)$$

where $L$ is the Laplacian matrix of the graph $G$. Then it follows from primal-dual optimality condition that, for $e = (u, v)$ we have

$$|f_{\Delta,T}^*(e)| = |x_{\Delta,T}^*(u) - x_{\Delta,T}^*(v)|.$$

Therefore, LF betweenness for all edges can be computed in time $\mathcal{O}\left(\lambda |V||E|\bar{d}^2 \log \frac{1}{\epsilon}\right)$. For sparse networks when $\bar{d}$ is constant, if we set $\lambda = \mathcal{O}(1/|V|) = \mathcal{O}(1/|E|)$, then the computation time reduces to $\mathcal{O}\left(|V| \log \frac{1}{\epsilon}\right)$. As a comparison, the computation time is at least $\mathcal{O}(|V|^2)$ for SP betweenness and $\mathcal{O}(|V|^2 \log |V|)$ for CF betweenness on sparse unweighted graphs. For arbitrary unweighted graphs, the time is $\mathcal{O}(|V||E|)$ for SP betweenness [54] and $\mathcal{O}(I(|V|) + |V||E| \log |V|)$ for CF betweenness [23], where $I(n)$ is the time to invert an $n \times n$ matrix. Note that small $\lambda$ is what we rely on to detect local contact bottlenecks (see Results). Therefore, for $\lambda$ relevant to our intervention method, computing LF betweenness can be several orders of magnitude faster than computing SP or CF betweenness (cf. Fig 12).

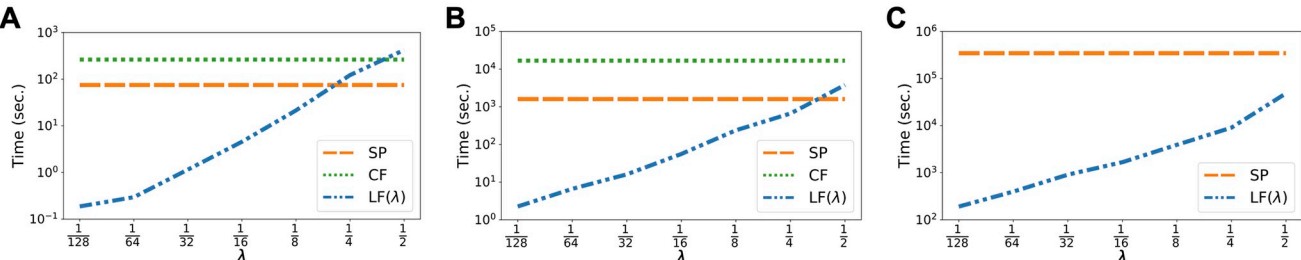

**Fig 12. Computation time of betweenness measures.** All computations are carried out on a personal laptop with 32GB RAM and 2.9 GHz 6-Core Intel Core i9. We used NetworkX [55] for computing SP and CF. We implemented LF computation in Julia. Note that the computation time scales linearly with λ. Moreover, since the range of λ values shown in the figures includes the values we used for our experiments, one may obtain accurate estimates on the computation times of all betweenness measures we used for the experiments. **(A)** Computation times for Facebook County. **(B)** Computation times for Port. Sub. **(C)** Computation times for Wi-Fi Montreal. CF is omitted for Wi-Fi Montreal because it takes too long to finish.

For completeness we layout a pseudocode for computing LF betweenness in Algorithm 1. The inner loop of Algorithm 1 is based on a randomized coordinate descent method that solves the dual problem (4) [26].

**Algorithm 1**. An efficient algorithm for computing LF betweenness

```
Input: An undirected graph G = (V, E). Degree vector d. Locality
parameter λ ∈ (0, 1]. Tolerance parameter ε > 0.
Output: The |V| × 1 LF betweenness vector betLF(λ) with parameter λ.
  b ← 0
  for s ∈ V do
    x ← 0
    r ← max{1_s - d/(λvol(G)), 0}, where max{a, 0} returns entry-wise
maximum
      while r(u) > ε for some u ∈ V do
        Pick any u ∈ V where r(u) > ε
        x(u) ← x(u) + r(u)/d(u)
        r(u) ← 0
        r(v) ← r(v) + r(u)/d(u) for each v ∈ V incident to u
      end while
      b(e) ← b(e) + |x(u) - x(v)| for each e = (u, v) ∈ E
  end for
  return b/|V|
```

## LF node-betweenness

Even though our definition of LF betweenness naturally applies to edges due to its physical interpretation of expected optimal flow in a diffusion process, one can trivially extend LF betweenness to quantify node importance, by aggregating flows on incident edges of a node. That is, we define the LF betweenness for a node $v \in V$ as

$$\mathbf{bet}^{\text{LF}}(v) \coloneqq \sum_{e \in E : v \in e} \mathbf{bet}^{\text{LF}}(e).$$

Note that the above relationship between edge-betweenness and node-betweenness applies to SP and CF betweenness as well.

## SEIR models

We use two different types of COVID-19 transmission models. Both assume an SEIR disease progression in the host where individuals are in one of four mutually exclusive compartments: susceptible to infection (S), infected but not yet infectious (E), infectious (I), and removed (R).

The first model described below is based on a system of ordinary differential equations [10] while the second is an agent-based model [20, 56, 57].

**ODE SEIR network model.**   Our ODE SEIR network model assumes that the proportion of susceptible, exposed, infectious and removed individuals in each population evolves according to an SEIR ODE model, and that transmission between populations occur through a network that connects these populations at rates determined by the network structure (connectivity and edge weights) of the Facebook County network. We define the following compartments:

- $S_i(t)$: number of susceptible persons at time $t$ in population $i$,

- $E_i(t)$: number of exposed persons (infected but not yet infectious) at time $t$ in population $i$,

- $I_i(t)$: number of infectious persons at time $t$ in population $i$,

- $R_i(t)$: number of removed persons at time $t$ in population $i$,

- $N_i$: number of persons in population $i$ (constant),

  and the following parameters:

- $A_{ji}$: the $(j, i)$th entry in the adjacency matrix of the Facebook County network, i.e., $A_{ji} = 1$ if there is an edge between population $j$ and population $i$. $A_{ji}$ captures edge weights in the network of populations, individuals in population $j$ can infect individuals in population $i$ as long as $A_{ji} > 0$,

- $\beta$: average transmission rate per unit time per contact,

- $\sigma_i$: average rate per unit time at which an individual transitions from the exposed stage to the infectious stage, in population $i$,

- $\gamma_i$: average rate per unit time at which an individual transitions from the infectious stage to the removed stage, in population $i$,

  The corresponding ODE SEIR network model is

$$\frac{dS_i}{dt} = -\beta \sum_j A_{ji} \frac{I_j}{N_j} S_i$$

$$\frac{dE_i}{dt} = \beta \sum_j A_{ji} \frac{I_j}{N_j} S_i - \sigma_i E_i$$

$$\frac{dI_i}{dt} = \sigma_i E_i - \gamma_i I_i$$

$$\frac{dR_i}{dt} = \gamma_i I_i \ .$$

For our simulations we assume $\sigma_i = \sigma$ and $\gamma_i = \gamma$ for all $i$. We did not vary the values of $\sigma_i$ and $\gamma_i$ across locations because there does not appear to be strong evidence that they vary across locations (even if they do vary with age, for instance) [58–60].

**Agent-based SEIR network model.**   To model infection spread in a network of individuals, we use an agent-based network SEIR simulation model [20, 56, 57]. An individual can be placed into one of following four states: (1) Susceptible (can contract the infection given contact with an infected individual), (2) Exposed (contracted the infection, but not yet infectious), (3) Infectious (with or without symptoms), and (4) Removed (either dead or obtained immunity and hence cannot infect others). The number of Susceptible, Exposed, Infectious,

Removed, and total individuals can be denoted as S, E, I, R, N, respectively. When an infectious individual passes the infection to a susceptible individual, the susceptible agent is activated. The algorithm allows us to keep track of the Exposed and Infectious agents over time. As the number of activated agents increases so does the computational expense. We assume that all edges have the same unit weight without intervention.

The total number of individuals within each of these disease states is given as:

- $S(t)$: number of susceptible persons at time $t$,

- $E(t)$: number of exposed persons (infected but not yet infectious) at time $t$,

- $I(t)$: number of infectious persons at time $t$,

- $R(t)$: number of removed persons at time $t$,

- $N$: number of persons in the population (constant),

  and the parameters are:

- $\beta$: transmission probability along a network edge, per unit time,

- $\sigma$: probability that a person transitions from exposed to infectious, per unit time,

- $\gamma$: probability that a person transitions from infectious to removed, per unit time.

  Each time step in the discrete-time simulation corresponds to one day. The corresponding algorithm is as follows

1. Loop over all nodes (each node is a person) for each time step. For each node, the following may happen

   - If a person is in state $S$, then each infected neighbouring person has a probability $\beta$ of infecting him/her, in which case the susceptible person moves from state $S \rightarrow E$.

   - If a person is in state $E$, s/he becomes infectious with probability $\sigma$ and the status changes from $E \rightarrow I$.

   - If a person is in state $I$, s/he recovers with probability $\gamma$ and the status changes from $I \rightarrow R$.

2. Update status of each person according to the events the person went through.

3. Repeat the steps for desired number of time steps.

## COVID-19 model parameterization

We set the average duration of the latent period $1/\sigma$ = 2.5 days and the average duration of the infectious period $1/\gamma$ = 5 days based on epidemiological data on COVID-19 serial interval and incubation period [61, 62]. (We note that the latent and infectious periods do not correspond to the incubation period and duration of illness [63].) We assumed a basic reproduction number $R_0$ = 2.5 for COVID-19 [64, 65]. We use the same values of $\sigma$ and $\gamma$ for both models. Calibration of the population-based ODE SEIR model for the Facebook County network and the agent-based SEIR model for the Wi-Fi Montreal and Portland networks required calibrating the value of $\beta$. In the agent-based network model, $\beta$ is simply the transmission probability per edge per time step. In the ODE model, $\beta$ is the coefficient of transmission in front of the adjacency matrix $A_{ji}$. In order to ensure comparability between these two model outputs, we calibrated their respective $\beta$ values to obtain the outcome that 85% of the population eventually becomes infected in the absence of any interventions, in both models (i.e., $\lim_{t\to\infty} \sum_i R_i/N_i = 0.85$). This

percentage was based on the final epidemic size on the Portland network when $\beta$ is set to match $R_0 = 2.5$ according to [36, 66]:

$$R_0 = \beta \frac{\langle k^2 \rangle - \langle k \rangle}{\langle k \rangle}, \tag{5}$$

where $\langle k \rangle$ and $\langle k^2 \rangle$ are the mean degree and the mean squared degree, respectively, of nodes in the network.

We used the Portland network to determine the target 85% final epidemic size for our experiments because the same parametrization for $\beta$ for the Facebook County network leads to unrealistic 100% final epidemic size while for Wi-Fi Montreal it leads to only 24% final size, which is too low for a pandemic. These irregular results for Facebook County and Wi-Fi Montreal using the parametrization (5) may be explained by the fact that Eq (5) holds under random graph assumption [66], however, the Wi-Fi Montreal is degree-irregular as majority of the nodes have degree 1; on the other hand, the epidemic process on the Facebook County network is simulated using population-based ODE model which may not behave exactly like agent-based network model that Eq (5) applies to. Because of these model limitations, in order to make sure that our experimental results are robust to different model parameterizations, we carried out additional experiments where we calibrated $\beta$ so that the final sizes are 70% and 55%, respectively, on each network (see Results).

We modelled edge weight reduction due to interventions by reducing $A_{ji}$ on the targeted edges for population-based ODE model and reducing $\beta$ values on the targeted edges for agent-based model accordingly.

## Intervention details

We assume all edges have weight 1. In order to simulate the effect of contact reduction on edges, once we have identified a set of edges for intervention, we reduce the corresponding edge weights by 90%. In the ODE model, this is implemented by setting $A_{ji} = 0.1$ if $(i, j)$ is an edge being targeted. In the agent-based model, this is implemented by setting $\beta_{ji} \leftarrow 0.1\beta_{ji}$ if $(i, j)$ is an edge being targeted. For all intervention strategies except UI, an $X$% coverage level means that we are targeting at the top $X$% of all edges at once according to the respective centrality measures. For UI, an $X$% interventional level means that all edge weights are reduced by $0.9X$%, so that the total amount of edge weight reduction in all interventions strategies are the same.

## Network community profile

In the seminal papers [24, 25] the authors studied how clustering structure of social networks changes as the size of the clusters increases. In particular, the NCP [24, 25] function is defined as:

$$\Phi(k) \coloneqq \min_{S \subset V} \phi(S) \text{ subject to : } |S| = k.$$

The NCP function takes as input the size $k$ and asks for the minimum conductance (cf. Eq (3)) that can be found in the graph such that the set $S$ has size $k$. The NCP can be used to calculate the clustering resolution profile of the network as the size of the set increases. Based on the NCP, many real-world networks can be classified into three distinct cases according to their "size-resolved community structure", (i) the best small communities have lower conductance than the best large communities (upward slopping NCP), (ii) the best small communities have comparable conductance to the best medium-sized and large communities (flat NCP), and

(iii) the best small communities have higher conductance than the best large groups (down-ward slopping NCP). Computing the NCP function is NP-hard and it cannot be computed exactly, it has been shown [24, 25] that NCP can be approximated (empirically) using local graph clustering algorithms [44, 49, 53, 67]. In our experiments we use the *Local Graph Clustering* API [39, 40] to approximately compute the NCP function for both original networks and the networks obtained from edge weight reduction due to intervention.

## Conclusion

Infection control methods that target features of network structure instead of features of individual nodes are increasingly feasible as empirical data on full contact networks becomes more abundant. At the same time, our network algorithms continue to improve. As we show here, LF betweenness computation can be orders of magnitude faster than SP or CF betweenness, which are the two most extensively exploited centrality measures for network epidemic interventions; moreover, our experiments show that physical distancing interventions based on LF betweenness mitigate a simulated COVID-19 epidemic on realistic contact networks more effectively than other network centrality based approaches. The superior computational efficiency and demonstrated intervention effectiveness make LF betweenness a more suitable candidate than others to improve more complex state-of-the-art network intervention strategies that rely on centrality measures. For example, dynamic interventions that sequentially remove nodes and edges or continuously vary edge weights could use LF betweenness to identify good targets. We see these methods working in tandem with digital contact tracing applications–such as COVID Alert–from which a network of contacts can be readily constructed and monitored. We suggest that public health control measures should evolve to reflect these new opportunities to improve pandemic mitigation.

## Supporting information

**S1 Text. CF betweenness is a special case of $\ell_2$-flow betweenness.**
(PDF)

**S2 Text. Experiment on LFR synthetic graph.**
(PDF)

**S1 Fig. Complete simulation results for the Facebook County network.** We illustrate predicted epidemic curves, epidemic peaks and epidemic sizes for four different initialization scenarios. We plot all epidemic curves at 25% intervention coverage level. We average over 50 trials for random initialization. Observe that targeting edges according to LF betweenness leads to the most reduction in epidemic sizes for all initialization scenarios, and it consistently lowers the epidemic peaks by a large amount. Furthermore, even though EG has a good performance in terms of total epidemic sizes, it almost has no effect in reducing the epidemic peaks, making EG an undesirable method. **(A)-(C)** The epidemic starts from New York, New York. **(D)-(F)** The epidemic starts from Los Angeles, California. **(G)-(I)** The epidemic starts from a well-connected cluster of 67 counties. **(J)-(L)** The epidemic starts from a random selection of 1% of all counties across the country.
(PDF)

**S2 Fig. Complete simulation results for the Wi-Fi Montreal network.** We illustrate predicted epidemic curves, epidemic peaks and epidemic sizes for two different initialization scenarios. We plot epidemic curves at 25% intervention coverage level. We average over 50 trials for random initialization. LF is shown to be the most effective at reducing both epidemic peaks and total outbreak sizes. **(A)-(C)** The epidemic starts from a well-connected cluster of 101

infected persons. **(D)-(F)** The epidemic starts from a random selection of 0.1% of all population as initially infectious.
(PDF)

**S3 Fig. Complete simulation results for the sub-sampled Portland network and the full Portland network.** We illustrate predicted epidemic curves, epidemic peaks and epidemic sizes for two different initialization scenarios. We plot epidemic curves at 25% intervention coverage level. We average over 50 trials for random initialization. **(A)-(C)** The epidemic starts from a well-connected cluster of 80 infected persons in the sub-sampled Portland network. **(D)-(F)** The epidemic starts from a random selection of 0.1% of all population in the sub-sampled Portland network as initially infectious. **(G)-(I)** The epidemic starts from a well-connected cluster of 37 infected persons in the full Portland network. **(J)-(L)** The epidemic starts from a random selection of 0.1% of all population in the full Portland network as initially infectious.
(PDF)

**S4 Fig. Simulation results when the transmission parameters are calibrated so that the final epidemic sizes on all datasets are 70%.** All other model parameters and intervention settings are the same. We show simulation results from random initialization of the SEIR models. Other model initializations lead to the same conclusion. LF intervention still delivers the best overall performance. **(A)-(B)** Results for Facebook County. **(C)-(D)** Results for Wi-Fi Montreal. **(E)-(F)** Results for sub-sampled Portland network. **(G)-(H)** Results for full Portland network.
(PDF)

**S5 Fig. Simulation results when interventions are applied in the middle of the pandemic instead of from the beginning.** All other model parameters and intervention settings are the same. We plot epidemic curves at 25% intervention coverage level to illustrate intervention effects that start in the middle. We show simulation results from random initialization of the SEIR models. Other model initializations lead to the same conclusion. LF intervention still delivers the best overall performance. **(A)-(C)** Results for Facebook County. **(D)-(F)** Results for Wi-Fi Montreal. **(G)-(I)** Results for sub-sampled Portland network. **(J)-(L)** Results for full Portland network.
(PDF)

**S6 Fig. Simulation results when reducing 99% weights instead of 90% on targeted edges.** All other model parameters and intervention settings are the same. We show simulation results from random initialization of the SEIR models. Other model initializations lead to the same conclusion. LF intervention still delivers the best overall performance. **(A)-(B)** Results for Facebook County. **(C)-(D)** Results for Wi-Fi Montreal. **(E)-(F)** Results for sub-sampled Portland network. **(G)-(H)** Results for full Portland network.
(PDF)

**S7 Fig. Simulation results for node immunization on Facebook County and Wi-Fi Montreal networks. (A)-(B)** Results for Facebook County. Node selection method based on LF betweenness gives the most reduction in both epidemic peaks and sizes, at all levels of node coverage. **(C)-(D)** Results for Wi-Fi Montreal. LF is the most effective when the node coverage is less than 10%.
(PDF)

**S8 Fig. Experiment on LFR synthetic graph. (A)** Percentage of "cut" edges targeted by different betweenness measures with varying percentage of targeted edges. **(B)** Final epidemic sizes

under different intervention strategies. We calibrated $\beta$ so that 85% of the population would be affected without any intervention. All other model parameters and intervention settings are the same. We used random initialization of the agent-based SEIR model and we averaged over 50 trials to obtain the results in (B). Observe the close relationship between the number of targeted "cut" edges and the final epidemic sizes: The more "cut" edges targeted, the more effective the intervention strategy is. Overall, LF is the most effective at identifying the "cut" edges, which in turn helps produce the most effective interventions.
(PDF)

**S9 Fig. Diffusion of initial mass with varying λ values.** In each plot, the diffusion starts from the single source node coloured in red. Edges that have a nonzero flow crossing them are coloured in yellow. The plots show that as λ increases, the initial mass spread further away from the source node. When λ = 1, the initial mass are diffused to every node in the graph.
(PDF)

## Acknowledgments

The authors are grateful to Thomas Hladish for providing the Wi-Fi Montreal network and to David F. Gleich for pointing to the Facebook County network.

## Author Contributions

**Conceptualization:** Shenghao Yang, Chris T. Bauch, Kimon Fountoulakis.

**Data curation:** Chris T. Bauch, Kimon Fountoulakis.

**Formal analysis:** Shenghao Yang, Kimon Fountoulakis.

**Methodology:** Shenghao Yang, Chris T. Bauch, Kimon Fountoulakis.

**Software:** Shenghao Yang.

**Supervision:** Kimon Fountoulakis.

**Writing – original draft:** Shenghao Yang, Priyabrata Senapati, Di Wang, Chris T. Bauch, Kimon Fountoulakis.

**Writing – review & editing:** Shenghao Yang, Di Wang, Chris T. Bauch, Kimon Fountoulakis.

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
