## [Decision Letter · Decision Letter 0]

17 Feb 2021

Dear Mr. Yang,

Thank you very much for submitting your manuscript "Targeted Pandemic Containment Through Identifying Local Contact Network Bottlenecks" for consideration at PLOS Computational Biology.

As with all papers reviewed by the journal, your manuscript was reviewed by members of the editorial board and by several independent reviewers. In light of the reviews (below this email), we would like to invite the resubmission of a significantly-revised version that takes into account the reviewers' comments.

We cannot make any decision about publication until we have seen the revised manuscript and your response to the reviewers' comments. Your revised manuscript is also likely to be sent to reviewers for further evaluation.

Sincerely,

Benjamin Muir Althouse

Associate Editor

PLOS Computational Biology

Stefano Allesina

Deputy Editor

PLOS Computational Biology

Reviewer's Responses to Questions

**Comments to the Authors:**

Reviewer #1: The review is uploaded as an attachment.

Reviewer #2: The authors present a novel measure (local-flow betweenness) for studying pandemic mitigation strategies in network models. They test compare the proposed measure to commonly-used measures via COVID-19 simulations on three real network data sets. Overall, the paper is well written and provides interesting methodology and results.

Comments:

* The authors mention computational considerations and the computational efficiency of the proposed LF over CF in several places in the manuscript (e.g., pg. 17). However, it would be beneficial to the reader if the authors included perhaps a table of CPU times to directly compare between the measures being tested in the simulation results.

* How is the optimal f*_{\\Delta,T} (pg. 15) computed in practice? More detail is warranted.

* The authors mention calibration of the transmission parameter \\beta (pgs. 21-22) in order to meet the assumption of R0=2.5 – how exactly was the calibration performed? How sensitive are the model outputs (and the overall results) on the value of this parameter and the other model parameters (\\sigma, \\gamma)?

* Can the authors comment on how reasonable is it to assume constant parameter values for average transmission, duration, etc., at different locations throughout the network in the SEIR ODE model (pg. 20)? Were there additional simulations run in which these values were assumed to vary based on location?

* Most of the figures are far too small to easily view / interpret without zooming in – consider making the plots larger in the final version of the manuscript for the ease of the reader.

**Have all data underlying the figures and results presented in the manuscript been provided?**

Reviewer #1: Yes

Reviewer #2: None

PLOS authors have the option to publish the peer review history of their article (what does this mean?). If published, this will include your full peer review and any attached files.

Reviewer #1: No

Reviewer #2: No
---

## [Decision Letter · Decision Letter 1]

27 May 2021

Dear Mr. Yang,

Thank you very much for submitting your manuscript "Targeted Pandemic Containment Through Identifying Local Contact Network Bottlenecks" for consideration at PLOS Computational Biology. As with all papers reviewed by the journal, your manuscript was reviewed by members of the editorial board and by several independent reviewers. The reviewers appreciated the attention to an important topic. Based on the reviews, we are likely to accept this manuscript for publication, providing that you modify the manuscript according to the review recommendations.

Sincerely,

Benjamin Muir Althouse

Associate Editor

PLOS Computational Biology

Stefano Allesina

Deputy Editor

PLOS Computational Biology

[LINK]

Reviewer's Responses to Questions

**Comments to the Authors:**

Reviewer #1: review is uploaded as an attachment

Reviewer #2: Thank you to the authors for their replies. I am left with a few more questions on the revised manuscript:

* In a reply to Review #1 regarding the optimal way to choose lambda in practice on real data, the authors suggest to perform a grid search and then “[s]imply pick the λ value to gives the best simulated intervention performance” (pg. 17) – grid searches are generally computationally expensive, can the authors comment on this / any limitations of choosing lambda in this manner? Further, it seems that the results are sensitive to the choice lambda – can this be further discussed? Are the authors suggesting to run the grid search on the real data sets directly, or on simulated data first? This is not clear in the newly added paragraph on pg. 17. If the latter is meant, how does this connect to the real data?

* I am still not convinced why the authors assume the values for sigma_i and gamma_i are fixed across all locations, as done in this study (see pg. 25, line 624) – can the authors comment on this? Were additional simulations run in which these values varied by location?

* (Minor) pg. 28, line 719: the term “local graph clustering” is written as one word (missing spaces).

**Have the authors made all data and (if applicable) computational code underlying the findings in their manuscript fully available?**

Reviewer #1: Yes

Reviewer #2: None

PLOS authors have the option to publish the peer review history of their article (what does this mean?). If published, this will include your full peer review and any attached files.

Reviewer #1: No

Reviewer #2: No

Figure Files:

Data Requirements:

Reproducibility:

References:

---

## [Decision Letter · Decision Letter 2]

13 Aug 2021

Dear Mr. Yang,

We are pleased to inform you that your manuscript 'Targeted Pandemic Containment Through Identifying Local Contact Network Bottlenecks' has been provisionally accepted for publication in PLOS Computational Biology.

Best regards,

Benjamin Muir Althouse

Associate Editor

PLOS Computational Biology

Stefano Allesina

Deputy Editor

PLOS Computational Biology

Reviewer's Responses to Questions

**Comments to the Authors:**

Reviewer #1: I am satisfied with the changes overall and I appreciate the clarifications in the method section. I recommend this manuscript to be accepted.

Reviewer #2: The authors have adequately addressed my concerns.

**Have the authors made all data and (if applicable) computational code underlying the findings in their manuscript fully available?**

Reviewer #1: Yes

Reviewer #2: None

PLOS authors have the option to publish the peer review history of their article (what does this mean?). If published, this will include your full peer review and any attached files.

Reviewer #1: No

Reviewer #2: No

---

## [Editor Report · Acceptance letter]

25 Aug 2021

PCOMPBIOL-D-20-02087R2 

Targeted Pandemic Containment Through Identifying Local Contact Network Bottlenecks

Dear Dr Yang,

I am pleased to inform you that your manuscript has been formally accepted for publication in PLOS Computational Biology. Your manuscript is now with our production department and you will be notified of the publication date in due course.

With kind regards,

Zsofi Zombor
